# Microstructure and Mechanical Properties of Ni-Based Alloy Composite Coating on Cr12MoV by Laser Cladding

**Yali Gao [1,\*], Yan Tong [1], Li Guohui [2], Pengyong Lu [1] and Dongdong Zhang [1,\*]**

1   Department of Mechanical Engineering, Northeast Electric Power University, No. 169 Changchun Road, Jilin 132012, China
2   Gongqing Institute of Science and Technology, Gongqing Shi, Jiujiang 332020, China
\*   Correspondence: 20102338@neepu.edu.cn (Y.G.); zhangdongdong@neepu.edu.cn (D.Z.)

**Abstract:** Cr12MoV has been widely used in the manufacture of stamping and drawing dies. In the present study, an attempt was made to improve the mechanical properties of Cr12MoV by laser cladding Ni60 alloy reinforced by WC. X-ray diffraction (XRD), scanning electron microscopy (SEM), a microhardness tester, and a friction and wear test prototype were used to analyze the macroscopic morphology, microstructure, and mechanical properties of the coating. The results showed that the coating mainly was composed of Cr-Fe-Ni, $\gamma$-(Fe, Ni), $Cr_{23}C_6$, $Cr_7C_3$, and $W_2C$ phases. The cladding layer presented the dendritic eutectic structure enriched Cr, Fe, and Ni. Zigzag-shaped dendrites with thicknesses of 50~80 μm of the bonding zone ensured the strong metallurgical bonding. Due to solid solution strengthening, dispersion hardening, and grain refinement, the hardness of the coating reached 745 HV, which was 3.5 times that of the substrate. The wear volume of the coating was $14 \times 10^{-3}$ mm$^3$, which was 48% lower than that of the substrate ($27 \times 10^{-3}$ mm$^3$). The coating had the abrasive wear; however, the substrate had the adhesive wear besides the abrasive wear.

**Keywords:** laser cladding; Cr12MoV; Ni-based alloy; microstructure; mechanical properties



## 1. Introduction

With the development of the industry, Cr12 series cold work die steel has been widely used in manufacturing industry due to its advantages of small deformation, high wear resistance, and large bearing capacity after heat treatment. Among them, Cr12MoV is the most widely used in this series [1]. Under working conditions, Cr12MoV often bears greater impact, extrusion, and external friction, and is prone to present failure forms such as wear and fatigue [2], which lead to potential safety hazards and great economic losses.

In order to improve the surface property of Cr12MoV, researchers have proposed and explored a variety of surface-treatment methods [3–7]. Laser surface treatment stands out among many surface-modification technologies because of its short processing time, flexible operation, and high accuracy [8]. In recent years, a series of laser surface treatments have been carried out on Cr12MoV to improve its performance [9–14].

At home and abroad, the cladding materials on Cr12MoV are mostly traditional alloys, such as Fe-based alloys, Co-based alloys, and Ni-based alloys. Although Fe-based alloys significantly improve the hardness and corrosion resistance, and reduce the cracking sensitivity of steel, the excessive Fe weakens the self-passivation ability and the anti-high-temperature oxidation properties of the coating [15,16]. Compared with Fe-based alloys, Co-based alloys [17–20] have better high-temperature mechanical properties and corrosion resistance. However, the higher cost of Co-based alloys has always been an obstacle that cannot be ignored.

Ni-based alloys [21,22] have good mechanical properties such as high hardness and strength. The hardness of pure Ni alloy, however, is insufficient, making it difficult to meet the performance requirements of cold stamping die. In order to further improve the

hardness and wear resistance of Ni-based alloy coatings, some researchers have adopted the methods of adding rare earth oxides or strengthening phases to reduce the friction effect and increase the service life of the material. Zhang Dongni [23] studied the influence of $CeO_2$ on the microstructure, hardness, and wear resistance of Ni60A-$Cr_3C_2$ coating. The results showed that after adding $CeO_2$ powder, the microstructure was significantly refined. When the addition amount of $CeO_2$ was 2 wt%, the coating achieved the largest microhardness (1107 HV).

Luo zhen et al. [24,25] added TiC and $Al_2O_3$ ceramic particles to reinforce Ni alloy. The results showed that the composite coatings had good metallurgical bonding with the substrate when the content of the ceramic particle was less than 15%, and TiC and $Al_2O_3$ were uniformly distributed in the solid solution, which significantly enhanced the wear resistance of the cladding layer. Yan Hua et al. [26,27] designed a Ni35/$MoS_2$/$LaF_3 \cdot CeF_3$ self-lubricating composite coating with a friction coefficient of 0.43 and a microhardness of 631 HV.

WC has been a research hotspot in enhancing Ni-based composite coatings owing to its high hardness and wear resistance, as well as its good wettability when combined with Ni. In literatures [28,29], Ni/WC composite coating was first applied as a laser-cladding material on Cr12MoV. It was found that the Ni60 + 30% WC coating had an optimal performance, and the wear resistance of the coating was approximately 30% higher than that of the coating without adding WC. Nevertheless, the hardness of the cladding layer was 67~68 HRC, only 30% higher than that of the substrate. In order to improve the coating hardness, Shen DaChen et al. [30,31] further explored and prepared a Ni60A + ω% WC (ω = 0, 15, 25, 35) gradient coating successfully. In this experimental conclusion, the average microhardness (82HRC) of the 35% WC coating reached 1.7 times that of the substrate. This indicates that different WC content has a great influence on the coating quality, and laser process parameters play a significant role as well. Therefore, Wang Ye [32] and Sun Wenqiang [33] et al. studied the effect of laser energy density on Ni/WC composite coating, which provides the data reference for the Cr12MoV repair process.

Based on the abovementioned research status, there are few studies about laser cladding Ni/WC composite coating on Cr12MoV at home and abroad, but the microstructure analysis has been not deeply carried out. Therefore, in this paper, the phase structure, microstructure solidification process, and the existence form of unmelted WC in laser cladding Ni/WC composite coating was studied in detail. However, the higher the content of WC is, the higher the hardness is. Excessive hardness results in poor plasticity and deformation defects. At the same time, excessive WC phase precipitating in the coating leads to the production of cracks. Therefore, in this study, WC content was reduced to 10% in order to obtain better strength and toughness. This also provides a theoretical basis for subsequent research.

## 2. Materials and Methods

### 2.1. Materials Used

Cr12MoV was selected as the substrate material in the present research; its chemical composition is listed in Table 1. The size of Cr12MoV is 50 mm × 30 mm × 10 mm. Prior to laser cladding, the surface was smoothed with 400 mesh sandpaper to give the surface some roughness and reduce laser reflectivity. The sample was then washed successively with acetone and alcohol to remove any oil spots.

**Table 1.** Chemical composition of Cr12MoV.

| Elements | Cr | C | Mo | V | Si | Mn |
|---|---|---|---|---|---|---|
| Mass percentage (wt%) | 11.0~12.5 | 1.4~1.70 | 0.4~0.6 | 0.2~0.3 | ≤0.4 | ≤0.4 |

The clad material was powdered Ni60/10% WC alloy, which was preplaced on Cr12MoV with a thickness of 1.0 mm. The chemical compositions of Ni60 are presented in Table 2.

**Table 2.** Chemical compositions of Ni60 alloy.

| Elements | Cr | B | Si | C | Fe | Ni |
|---|---|---|---|---|---|---|
| Mass percentage (wt%) | 14~17 | 2.5~4.5 | 3~4.5 | 0.6~0.9 | ≤15 | Bal. |

### 2.2. Laser Machining

The experiment set a DL-2000 cross-flow $CO_2$ laser (Shenyang Continental Laser Complete Equipment Co., LTD, Shenyang, China) as the heat source with a maximum power of 2 kW. Its working principle is shown in Figure 1. The prefabricated powder method was used before laser cladding. The thickness of the powder was determined by a slide with a size of 50 mm × 30 mm × 1 mm. The size of the middle groove of the slide is 45 mm × 25 mm × 1 mm. The powder was filled into the groove to obtain a powder with a height of 1 mm. The optimized parameters of laser cladding were obtained by orthogonal test as follows: laser power of 1.5 kW, scanning speed of 200 mm/min, spot diameter of 3 mm, overlap rate of 30%, and argon flow rate of 5 L/min.

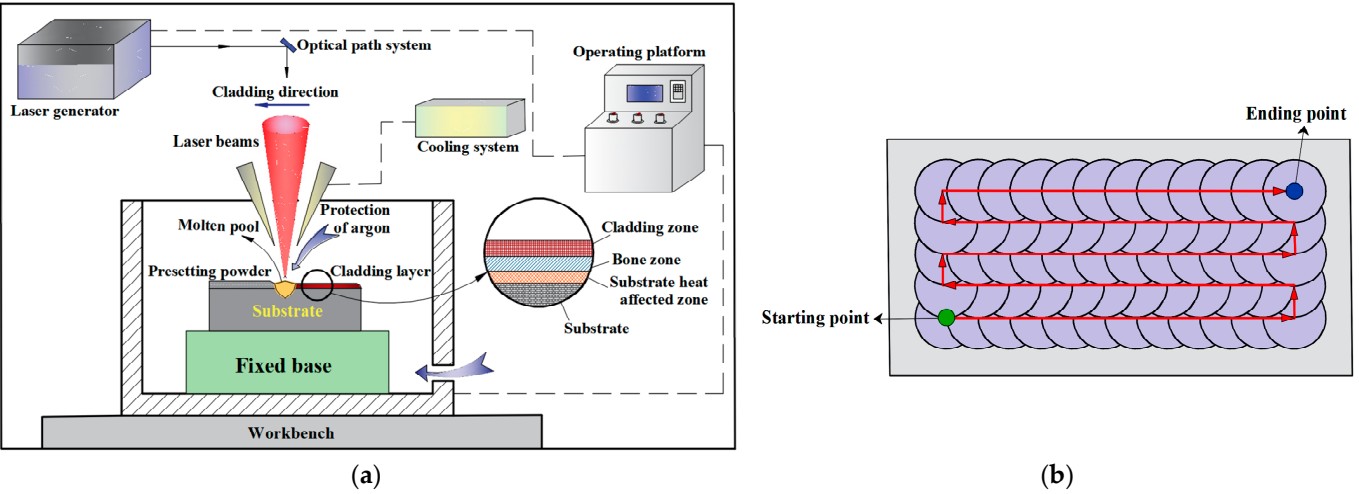

(**a**)　　　　　　　　　　　　　　　　　　　　　　　　　　　(**b**)

**Figure 1.** Principle diagram of the laser-cladding process: (**a**) laser processing process; (**b**) scanning path.

### 2.3. Microstructure and Properties Analysis

TD-3500 X-ray diffraction (Dandong Tongda Technology Co., LTD, Dandong, China) was used to analyze the coating phase in detail. The instrument was equipped with a Ni-filtered, Cu Ka source operating at 40 kV and 30 mA. The data were collected in the range 20°~80° with a step size of 0.028 and step time of 0.3 s; the experimental time was 30 min and the scanning speed was 2°/min.

Microstructural characterization of the coating was observed and analyzed using JSM-7610s SEM (Shanghai Baihe Instrument Technology Co., LTD, Shanghai, China). Samples used for SEM were etched using aqua regia solution.

Microhardness of the cross section of coatings was measured by a HXD-1000TMC/LCD (Wuxi Metes Precision Technology Co., LTD, Wuxi, China) Vickers tester with a test load of 200 g and loading time of 15 s. Sample surfaces were smoothed and polished with 1200 mesh sandpaper and tested for microhardness at 3 points per cross section. The samples with 10 mm × 10 mm × 5 mm were cut from the as-received Cr12MoV and laser cladding samples for wear studies, and ground on 800 grit size emery paper to obtain the same surface finish. The measurement of wear resistance was performed by MGW-02

(Jinan Yihua Tribology Testing Technology Co., LTD, Jinan, China) wear testers with 5 mm diameter steel ball bearings; the hardness of the steel ball was HRC58, and the test temperature and relative humidity were $25 \pm 1\ °C$ and 60%, respectively. The testing parameters were load of 5 N, frequency of 10 Hz, sliding time of 20 min, and reciprocal sliding distance of 3 mm. Figure 2 shows the schematic diagram of the friction-wear test.

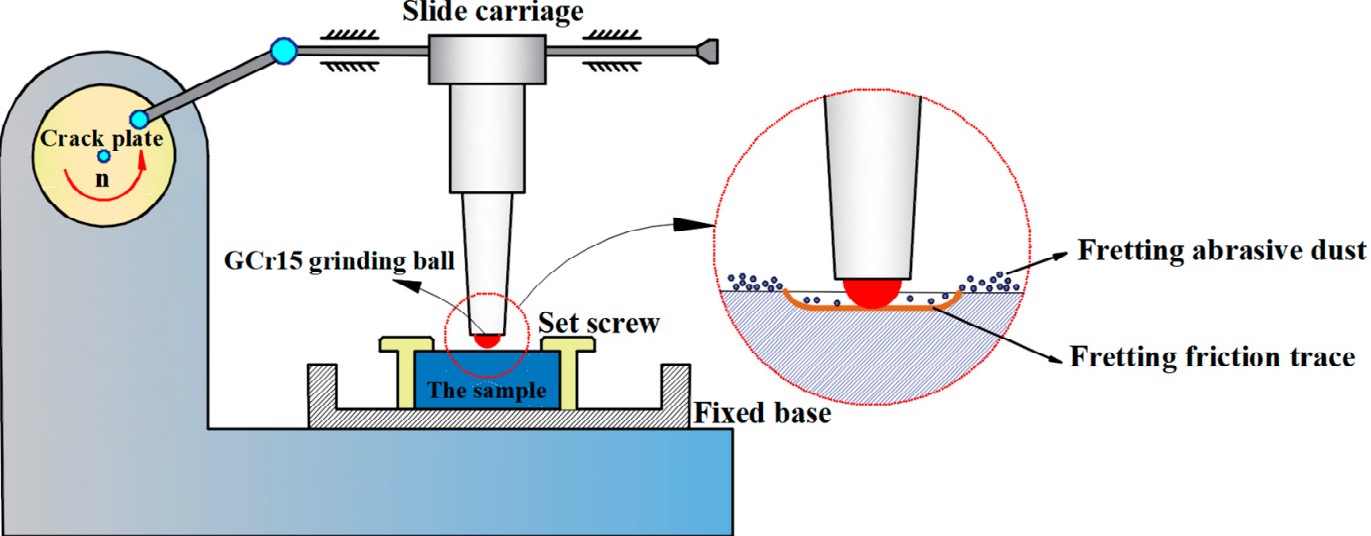

**Figure 2.** Schematic diagram of the friction and wear experiment.

Figure 3 displays the calculation principle of wear volume. In Figure 3, the circle with radius $R$ is the section along the diameter of the grinding ball, $L_{AB}$ is the width of the grinding mark, $2\theta$ is the central angle corresponding to the width of the grinding mark, $S_1$ is the area of the sector, $S_2$ is the area of the triangle contained in the sector, and $S_3$ is the area of the grinding mark section. The first step is to solve the central half, angle theta $\theta$:

$$\theta = \arcsin(L_{AB}/2R) \tag{1}$$

then find the sector area $S_1$ corresponding to arc AB:

$$S_1 = 2\theta\pi R^2/2\pi = \theta R^2 \tag{2}$$

and find the area $S_2$ of $\Delta_{OAB}$:

$$S_2 = \frac{1}{2}L_{AB} \cdot R \cdot \cos\theta \tag{3}$$

From the above, the wear mark section area $S_3$ is obtained:

$$S_3 = S_1 - S_2 \tag{4}$$

Finally, the wear volume $V$ is obtained:

$$V = S_3 \times L_{AB} \tag{5}$$

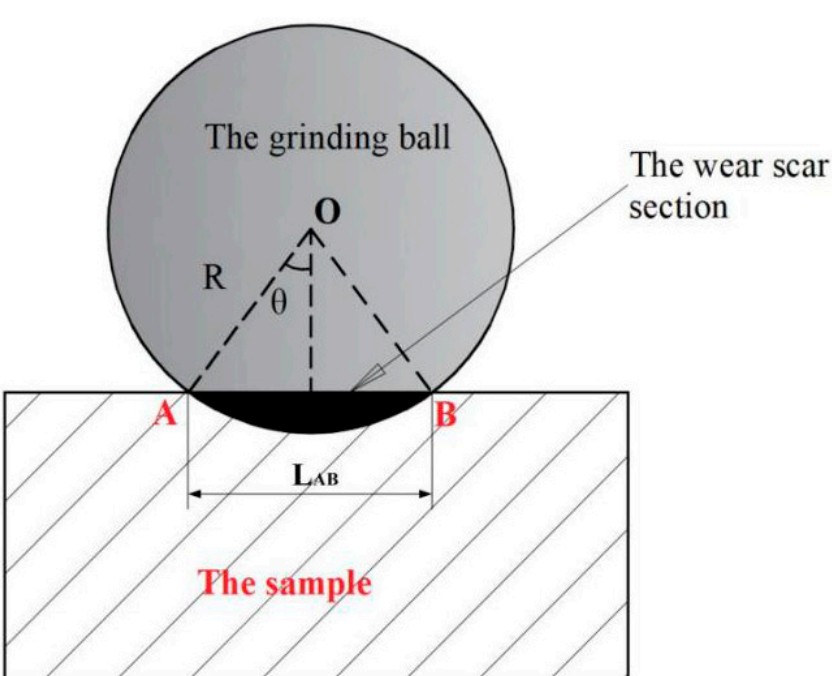

**Figure 3.** Schematic diagram of the wear volume calculation. $O$ is the center of the ball; $R$ is the radius of the sphere; $\theta$ is the center half angle corresponding to the width of the grinding mark; $L_{AB}$ is the width of the grinding mark.

## 3. Results and Discussion

### 3.1. Macromophology of the Coating

The coating macromophology with power of 1.5 kW, scanning speed of 200 mm/min is shown in Figure 4. It can be seen that the coating appears to be a dense arrangement of fish scales, orderly, smooth, and continuous, with no obvious pores, cracks, or other defects.

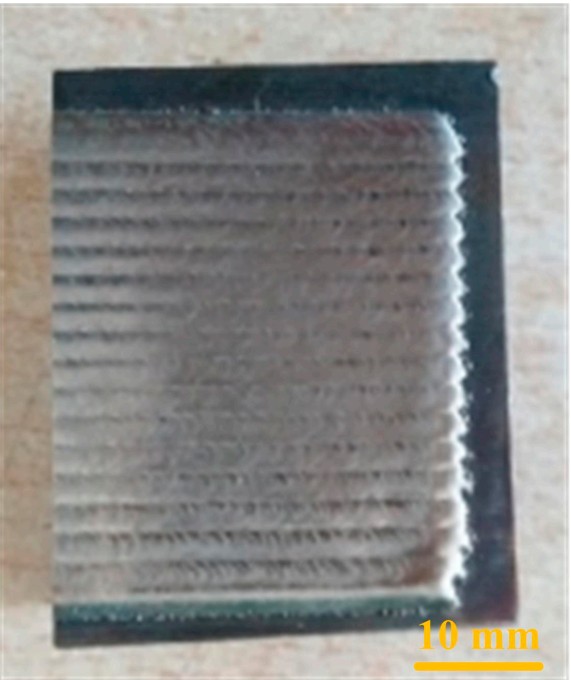

**Figure 4.** Macromophology of the coating.

*3.2. Microstructure Analysis of the Coating*

3.2.1. XRD Analysis of the Coating

The XRD analysis result of the coating is shown in Figure 5. The coating is mainly composed of Cr–Fe–Ni, $\gamma$-(Fe, Ni), $Cr_{23}C_6$, $Cr_7C_3$, and $W_2C$ phases. Under the action of high temperature of laser cladding, WC firstly decomposed to form $W_2C$ in situ. Then, the precipitated C and other alloying elements formed a variety of carbides, such as $Cr_{23}C_6$ and $Cr_7C_3$. These hard phases play a dispersively strengthened role in the coating. Finally, $\gamma$-(Fe, Ni) solid solution was formed by the solution of Fe into $\gamma$-Ni, and $\gamma$-(Fe, Ni) solid solution formed a complex Cr–Fe–Ni phase with Cr. The combined action of solid-solution strengthening and the carbide hard phase obviously improved the microhardness and wear resistance of Cr12MoV, and reduced the probability of scrap due to friction and wear in the process of using the die.

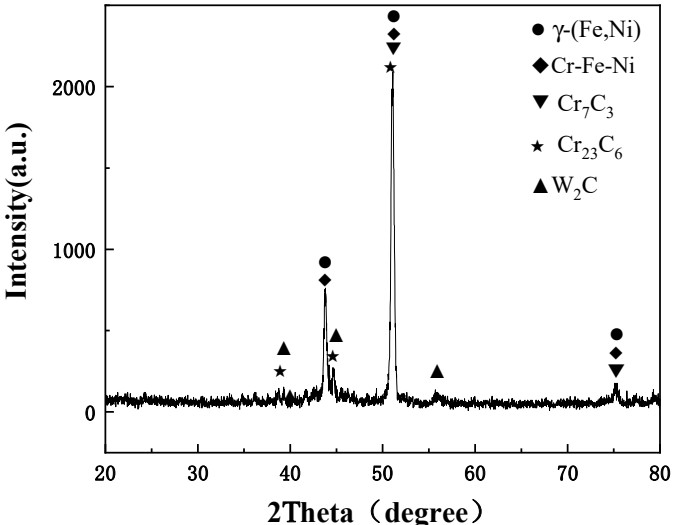

**Figure 5.** XRD analysis results of the coating.

3.2.2. Analysis of Microstructure of the Coating

Figure 6 shows the microstructure of the coating. From Figure 6a, it can be seen that the coating was mainly composed of the cladding layer, bonding zone, and substrate. It is clear that the interface between the cladding layer and substrate presented good metallurgical bonding without obvious defects such as holes and cracks. Figure 6b shows the line scan result of the coating. It can be observed that Fe, Ni, Cr, and W had an obvious mutual diffusion in the bonding zone, which ensures the bonding strength of the coating and the substrate.

The magnified microstructure of the cladding layer is shown in Figure 6c, which indicates a dendrite structure. EDS analysis was carried out on the dendrite (1, 3 points) and the inter-dendrite (2, 4 points) respectively. The results are listed in Table 3. It can be seen that the dendrite was mainly rich in Cr, Fe, Ni, and C. Compared with "2" and "4" points, there were more Cr elements at points "1" and "3". This is because during the cooling and solidification process, Cr precipitates from inter-dendrite to dendrite. At the same time, Cr and C generate new carbides. Combined with the XRD results, it can be inferred that the dendrite is a eutectic structure formed by $Cr_7C_3$ and Cr–Fe–Ni. Furthermore, a large amount of Ni was found in the inter-dendrite, which shows that the inter-dendrite was $\gamma$-(Fe, Ni) solid solution. The results are consistent with the analysis results in Figure 5.

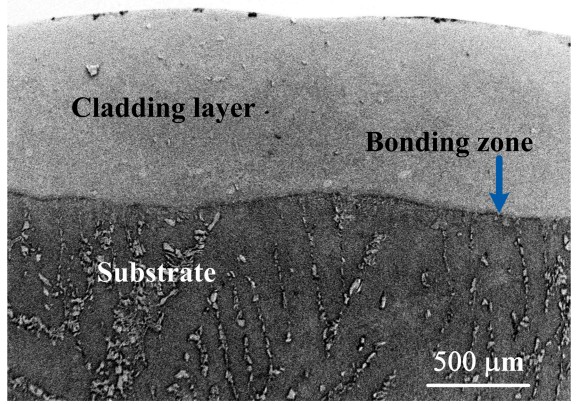

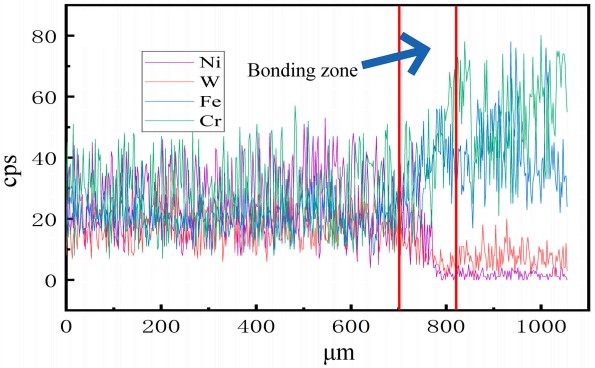

(**a**)  (**b**)

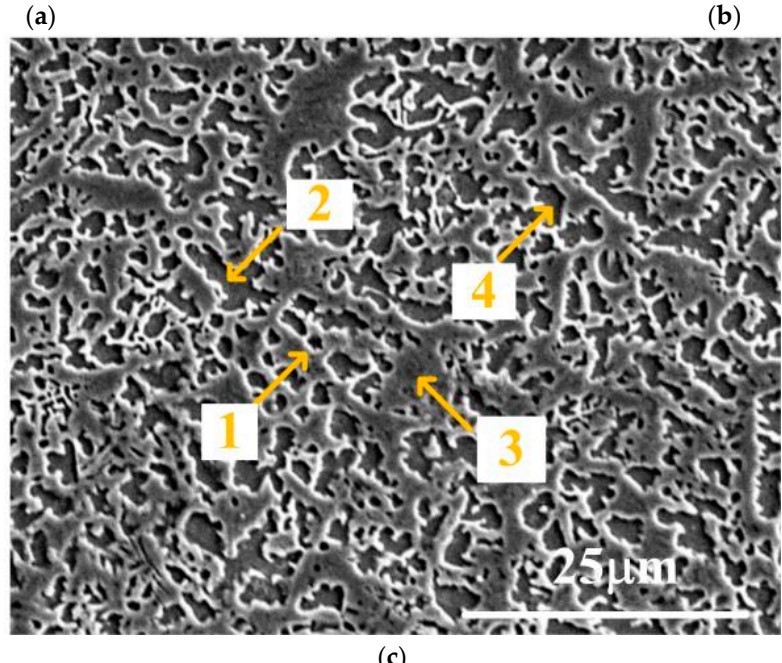

(**c**)

**Figure 6.** The microstructure of the coating. (**a**) Cross-sectional morphology of the coating. The dendrite (1, 3 points) and the inter-dendrite (2, 4 points); (**b**) result of the line scan; (**c**) magnified microstructure of the cladding layer.

**Table 3.** EDS results of different positions in the cladding layer (at%).

| Position | C | V | Cr | Fe | Ni | Mo | W |
|---|---|---|---|---|---|---|---|
| 1 | 20.37 | 0.28 | 41.60 | 19.76 | 17.05 | 0.00 | 0.93 |
| 2 | 24.83 | 0.07 | 7.06 | 19.41 | 47.45 | 0.00 | 1.18 |
| 3 | 31.11 | 0.05 | 32.82 | 13.34 | 21.68 | 0.00 | 1.01 |
| 4 | 23.75 | 0.04 | 8.91 | 19.48 | 46.72 | 0.00 | 1.10 |

### 3.2.3. Analysis of Dendrite Change with the Increasing of the Layer Depth

Figure 7a–c show the dendrites of the top, middle, and bottom parts of the coating respectively. It is obvious that with the increasing of the coating depth, the dendrite became gradually coarser. Because laser cladding is a process of rapid heating and solidification, the temperature gradient (G) gradually decreased and the cooling rate gradually increased from the bottom to the upper part of the molten pool, which led to the dendrite becoming finer.

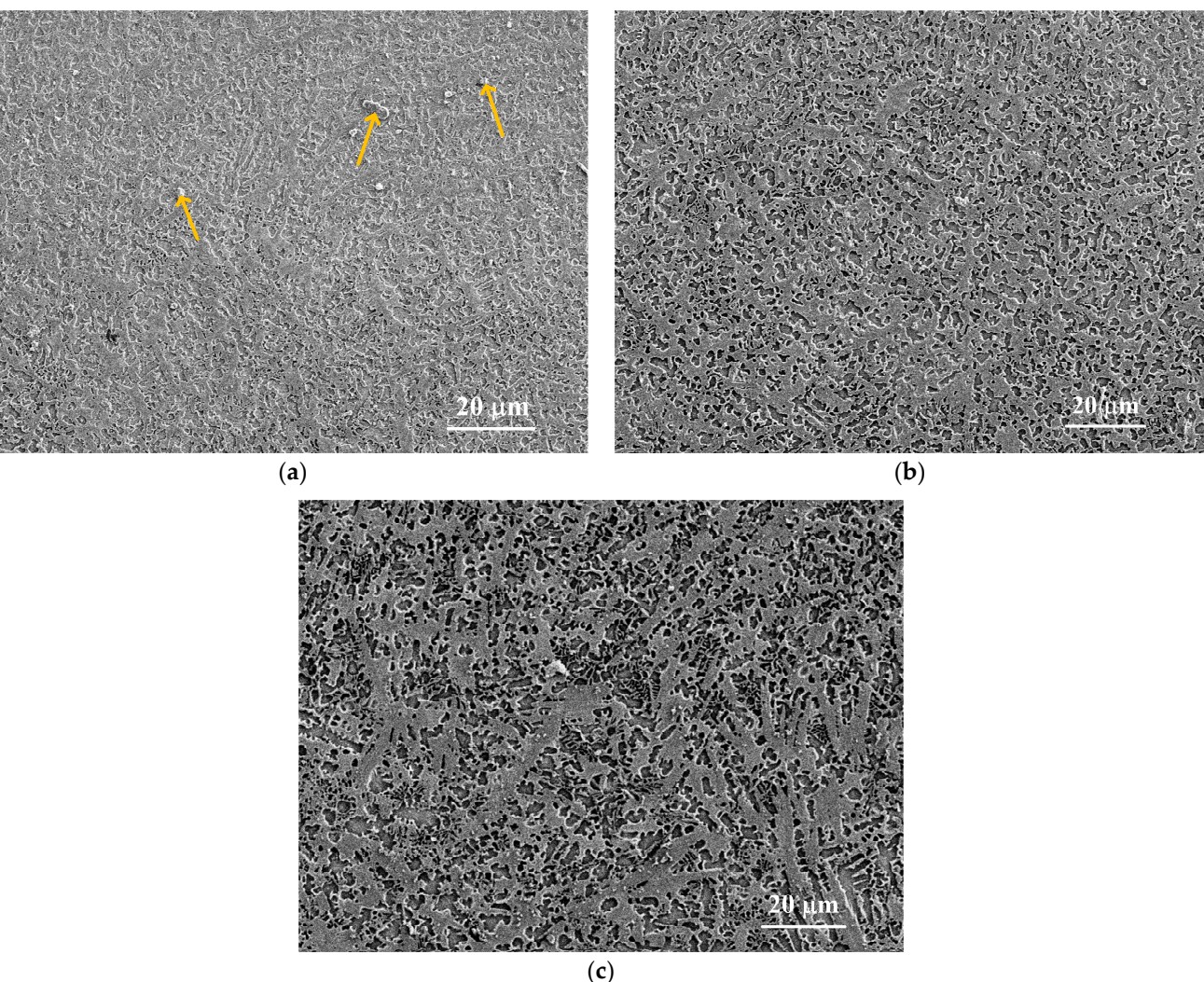

**Figure 7.** The dendrite morphology of the coating from top to bottom: (**a**) dendrite morphology in the top of the coating (Yellow arrows indicate particles at the top of the coating); (**b**) dendrite morphology in the middle of the coating; (**c**) dendrite morphology in the bottom of the coating.

Moreover, it can be observed from Figure 7a that there are many particles on the top of the coating (indicated by the yellow arrows). EDS results showed that the content of C (45.27%) was relatively high in the particles (Table 4). According to the solidification theory of metals, carbide in the molten pool underwent the following reactions [34]:

$$L + WC \rightarrow W_2C \tag{6}$$

$$L + (Cr, Fe)_2C \rightarrow (Cr, Fe)_7C_3, \tag{7}$$

$$L + (Cr, Fe)C \rightarrow (Cr, Fe)_{23}C_6 \tag{8}$$

**Table 4.** Elemental composition of surface particles (at%).

| Element | B | C | Si | Cr | Fe | Ni | W |
|---|---|---|---|---|---|---|---|
| mass percentage | 0.00 | 45.27 | 0.09 | 25.69 | 28.89 | 0.00 | 0.06 |

Together with the results of XRD analysis, it can be deduced that the particles are composed of interstitial compounds such as $Cr_{23}C_6$.

### 3.2.4. Analysis of Bulk Hard Phase in the Bottom of the Coating

Figure 8 exhibits the bulk hard phase in the bottom of the coating. The bulk phases dispersively distributed at the bottom of the molten pool (referred to in the box in Figure 8a, and the magnified morphology is shown in Figure 8b). It is obvious that the bulk phase was closely embedded in the dendrite, and white granular material was distributed on its surface. In order to further determine the phase composition, surface scanning analysis was carried out. The results are shown in Figure 9.

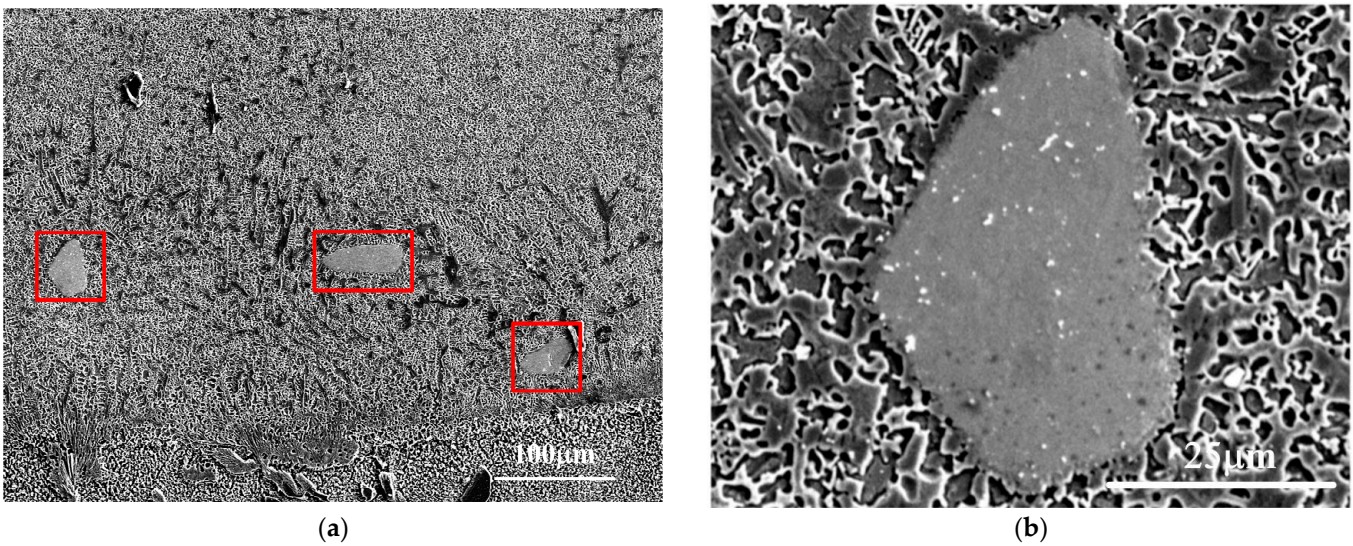

(**a**)  (**b**)

**Figure 8.** Bulk hard phase in the bottom of the coating: (**a**) distribution of the bulk hard phase (Red squares indicate the hard phase at the bottom of the coating); (**b**) magnified bulk hard phase.

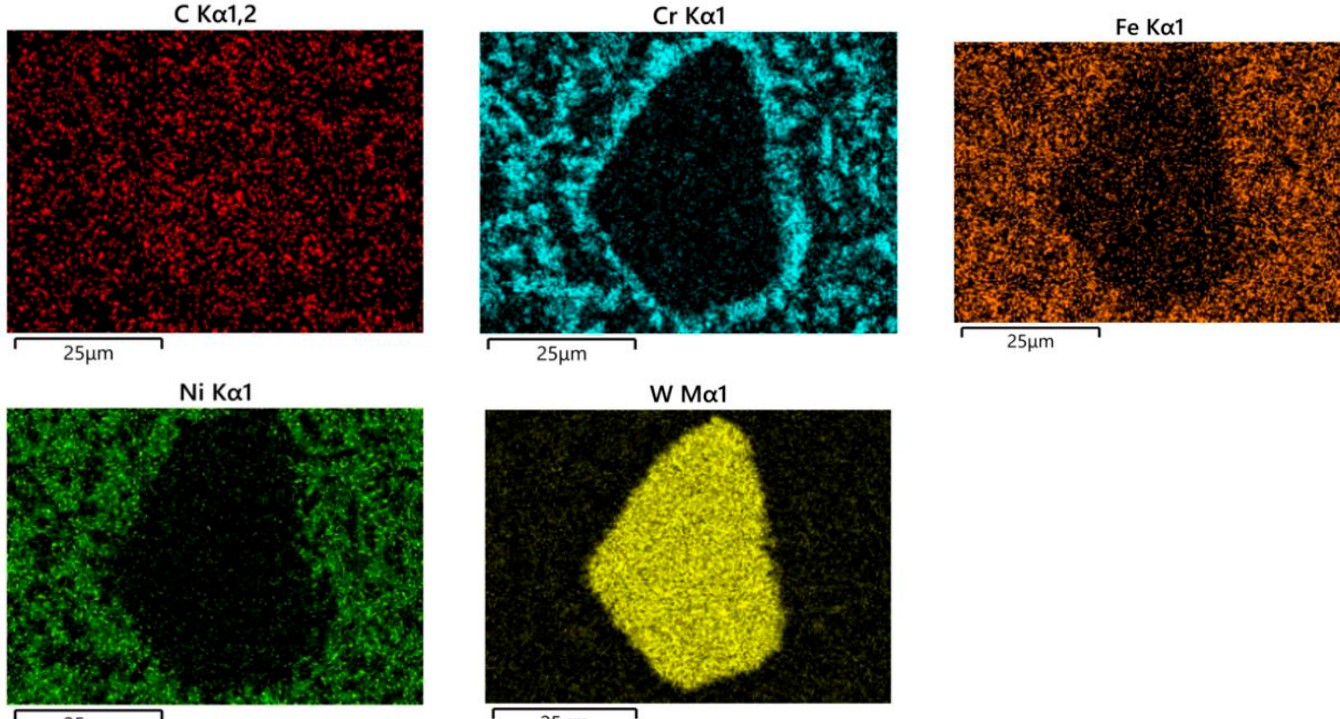

**Figure 9.** Results of face scanning.

It is deduced that the bulk phase is unmelted WC from Figure 9. During the solidification process of the molten pool, unmelted WC was distributed at the bottom of the molten

pool owing to its large density, which provides the core of non-uniform nucleation. Therefore, the nickel-based eutectic structure takes it as the core for epitaxial growth (Figure 8b), and the hardness test showed that the hardness of bulk WC was 1293 HV. The dispersion distribution of a hard phase with high hardness at the interface can significantly improve the interface strength.

### 3.2.5. Analysis of Zigzag Microstructure of the Binding Zone

Figure 10 shows the microstructure of the binding zone. The dense interface structure indicates that the coating formed good metallurgical bonding with the substrate. This is because the substrate surface was polished by coarse sandpaper before laser cladding in order to increase the bonding strength, and the surface of the substrate is uneven, which led to the inconsistency of the depth of the molten pool. After solidification, the zigzag-shaped dendrite grew along the vertical interface direction with the thickness range of 50~80 μm.

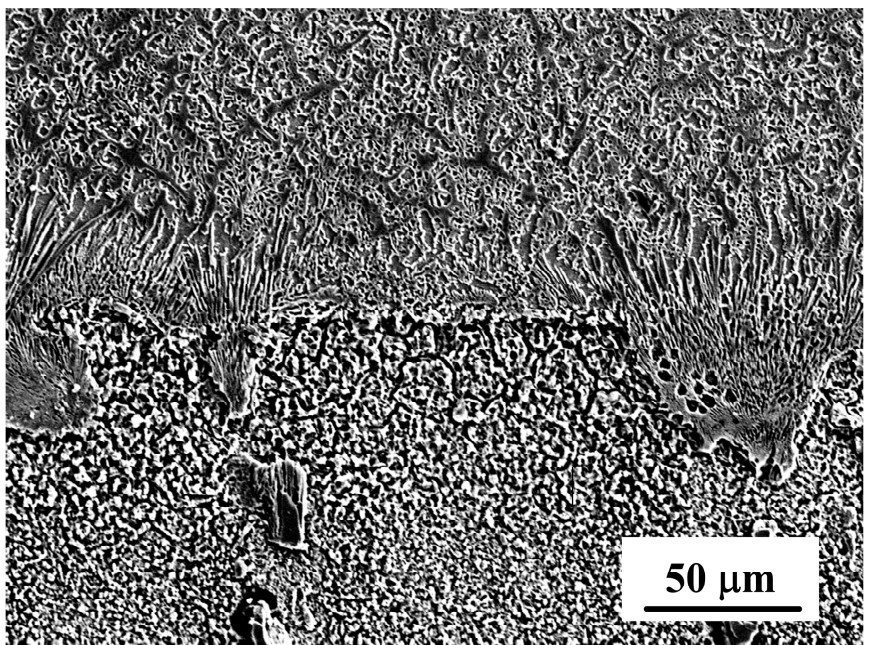

**Figure 10.** The microstructure of the binding zone.

### 3.3. Properties Analysis of the Coating

### 3.3.1. Analysis of the Hardness of the Coating

The microhardness curve of the coating is shown in Figure 11. The average hardness of the coating (515 HV) was 59% higher than that of the substrate (212 HV). There exists the presence of supersaturated Cr, Fe, and C in the solid solution formed by the solidification of the coating. These elements play a role in solid solution strengthening. The hard phases formed by C and Cr, Fe, and W play a diffusion-hardening role. In addition, grain refinement in the coating also improves the hardness.

On most of the surface of the coating, the microhardness was relatively stable in 550 HV. Microhardness at the secondary surface reached the maximum value of 745 HV (0.3 mm from the coating surface), which is about 3.5 times that of the substrate. This is because in the process of laser cladding, the surface elements are burned, resulting in the hard phase content being lower than that of the secondary surface. Additionally, the impurities in the molten pool rose during the cooling and solidification process. It created a loose coating on the surface, which reduced the hardness of the coating surface.

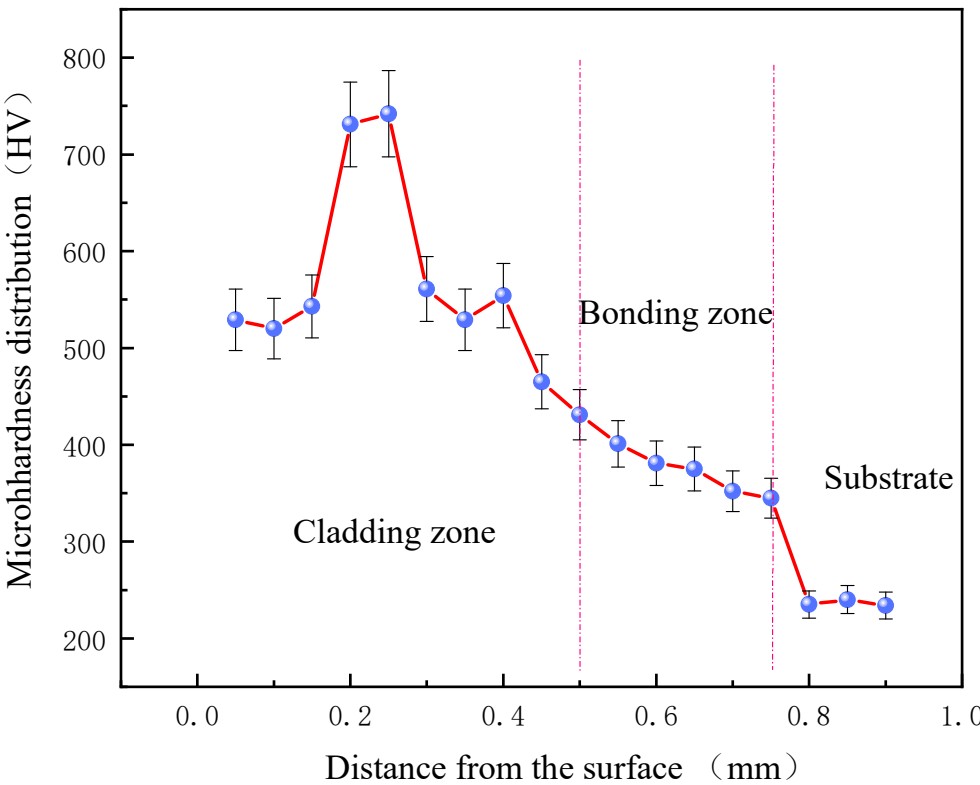

**Figure 11.** The microhardness curve of the coating.

### 3.3.2. Analysis of the Wear Resistance of the Coating

Figure 12 shows the friction coefficient curves of the coating and substrate. The friction coefficient of the coating and substrate was 0.08 and 0.12, respectively. The average friction coefficient of the coating was 33% lower than that of the substrate during the period of 0–20 min. This is because Ni60 alloy has good self-lubrication, and in the friction and wear process it has good friction reduction compared with the substrate.

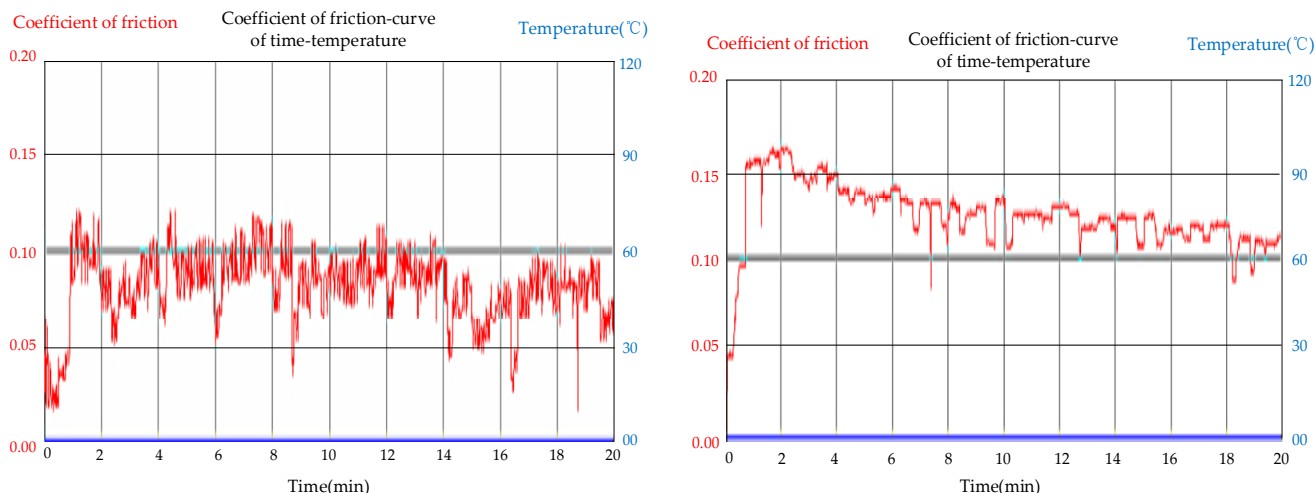

**Figure 12.** The friction coefficient curves of the coating and the substrate: (**a**) the coating; (**b**) Cr12MoV substrate.

Figure 13 shows the wear morphology of the substrate. It can be seen that the wear surface of the substrate was rough and had obvious furrows and debris, which indicates that the wear mechanism was adhesive wear and abrasive wear.

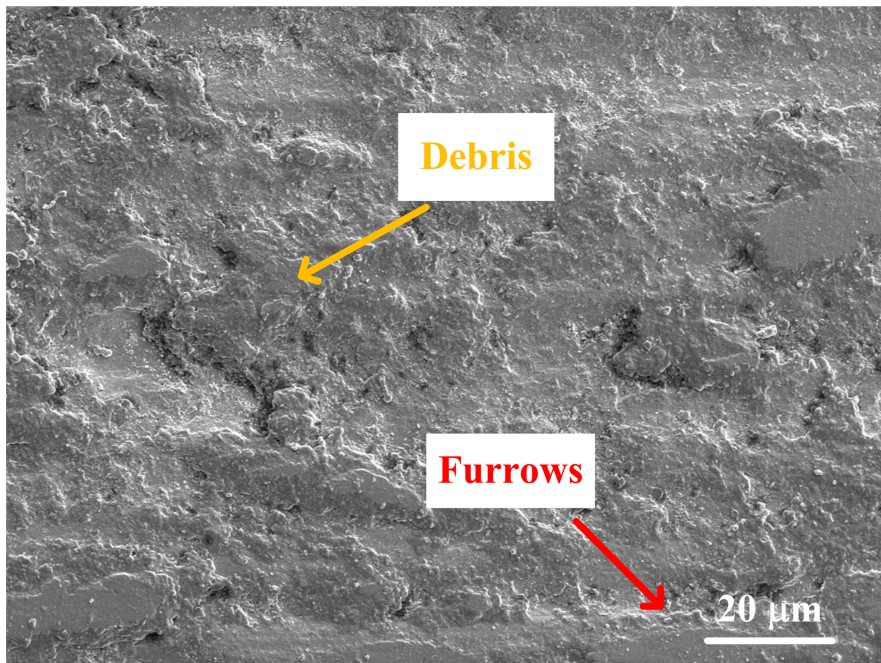

**Figure 13.** Wear morphology of the substrate.

In the sliding wear process, the hardness of the substrate was far less than that of GCr15. With the increase of sliding time, the GCr15 ball was repeatedly chipped on the surface of the substrate, so a lot of debris fell off from the substrate. Those falling debris were not excluded in time, and were rolled into a lamellar structure under the action of loading force. EDS analysis (Table 5) showed that lamellar debris contained a large amount of Fe and O; therefore, it can be concluded that the sliding process was accompanied by the occurrence of an oxidation phenomenon.

**Table 5.** EDS results of lamellar debris in the substrate (at%).

| Element | O | Si | Cr | Fe | Mo |
|---------|------|------|------|-------|------|
| At% | 22.00 | 0.88 | 9.02 | 67.93 | 0.18 |

The coating only had the abrasive wear shown in Figure 14. The wear surface of the coating was relatively flat, and there was no obvious plastic deformation. Because of the high hardness, the coating effectively resisted the plastic deformation.

Through calculation, the wear volume of the coating was found to be $14 \times 10^{-3}$ mm$^3$, and the wear volume of the substrate was $27 \times 10^{-3}$ mm$^3$. The wear resistance of the coating is increased to 48% compared with the substrate. Under the abrasive wear mechanism, the wear resistance of the material is directly proportional to its hardness. Moreover, the coating had significantly higher hardness than the substrate, so the coating exhibited lower wear volume and higher wear resistance.

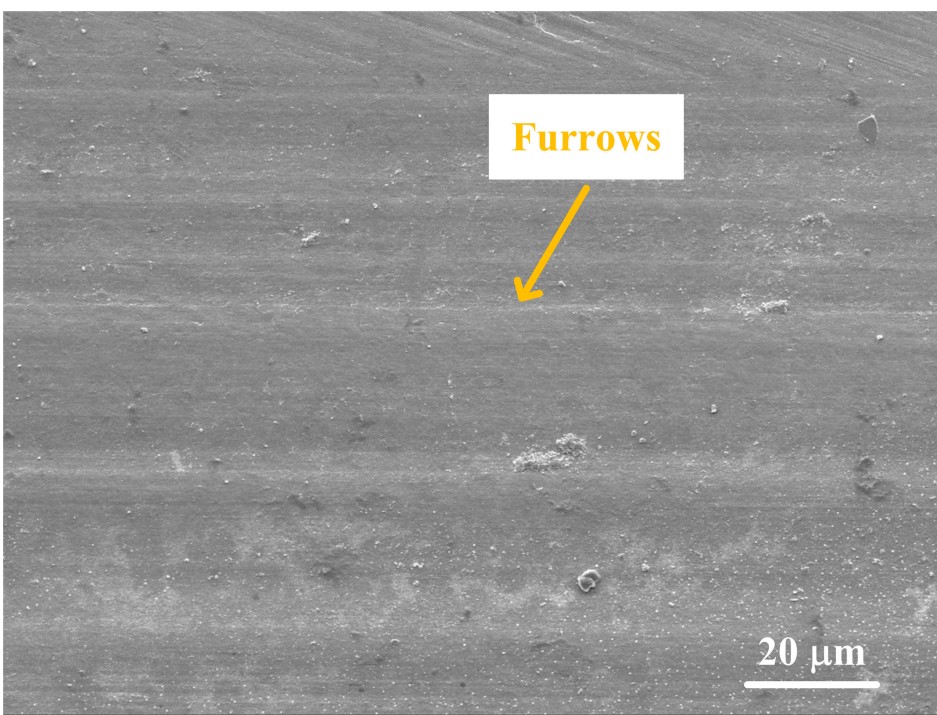

**Figure 14.** Wear morphology of the coating.

## 4. Conclusions

(1) In this study, an attempt was made to improve the mechanical properties of Cr12MoV by laser cladding Ni60 alloy reinforced by WC. The results showed that the hardness and wear resistance of the coating were significantly improved compared with that of the substrate. The details are as follows:

(2) The phases of the coating are composed of Cr–Fe–Ni, $\gamma$-(Fe, Ni), Cr23C6, Cr7C3, and W2C. The microstructure of the coating layer is dendrite. The supercooling degree from the bottom to the top of the molten pool gradually increased, resulting in the dendrite morphology changing from coarse to fine.

(3) The zigzag microstructure of the bonding zone ensured the metallurgical bonding strength. The undisolved WC diffusive distributed in the bottom of the coating, which significantly improved the interfacial bond strength. In addition, WC particles were tightly embedded in Ni dendrites, which reduced the cracking sensitivity of the coating. There were no cracks, holes, or other defects in the interface between the cladding layer and substrate.

(4) The average hardness of the coating was 59% higher than that of the substrate under the combined effects of solution strengthening, diffusion hardening, and fine grain strengthening. Because Ni60 alloy has good self-lubricity, the friction coefficient of coating was 33% lower than that of the substrate. The wear mechanism of the coating was mainly abrasive wear, and the wear mechanism of the substrate was adhesive wear and abrasive wear. Under the abrasive wear mechanism, the wear resistance of the material was proportional to the hardness, so the coating had a high wear resistance. Compared with the substrate, the wear volume of the coating was reduced by 48%.

**Author Contributions:** Formal analysis, Y.G. and D.Z.; data curation, P.L.; writing—original draft preparation, L.G. and Y.T. All authors have read and agreed to the published version of the manuscript.

**Funding:** This research was funded by "The 13th Five-Year" Science and Technology research in Jilin Province and the Project of Science and Technology Development of science and Technology Bureau in Jilin, grant number "20220098KJ" and "201831785".

**Institutional Review Board Statement:** Not applicable.

**Informed Consent Statement:** Not applicable.

**Data Availability Statement:** Not applicable.

**Acknowledgments:** Special thanks to the laser Laboratory of Northeast Electric Power University for the help of this study.

**Conflicts of Interest:** The authors declare no conflict of interest. The funders had no role in the design of the study; in the collection, analyses, or interpretation of data; in the writing of the manuscript; or in the decision to publish the results.

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
