# Peer review of "Microstructure and Mechanical Properties of Ni-Based Alloy Composite Coating on Cr12MoV by Laser Cladding"

_coatings, doi:10.3390/coatings12111632_

Round 1
Reviewer 1 Report
The article topic is relevant. However, there are too many deficiencies. Here are some specific remarks:
1. Abstract: It does not fully reflect the content and summarize the problem, the method, the results, and the conclusions; therefore, it should be improved.
2. It is necessary to clearly show in the introduction what is the novelty and originality of this work.
3. It is necessary to provide the actual, experimentally determined chemical composition of Cr12MoV steel and Ni60 alloy.
4. The brands of the equipment used (X-ray diffractometer, scanning electron microscope, microhardness tester) should be provided. For the diffractometer, it is also necessary to provide the radiation used. How the wear volume was calculated?
5. Some questions are raised by diffractogram (Figure 4) and EDS analysis (Figure 5c, Table 3). How do the authors understand the difference between Cr-Fe-Ni and γ-(Fe, Ni)? This must be proved by other methods. In my opinion, the phase composition of the coating is determined incorrectly. It can be expected that the structure consists of a Ni-based solid solution, eutectic Ni+? (the composition of the eutectic should be clarified) and different strengthening phases. Points 1 and 3 shown in Figure 5c are most probably chromium carbides. This part of the article should be fully revised.
6. Figures 1a; 2; 5c; 11: The text inscriptions are barely visible. This should be corrected.
7. Figures 3; 5b,c: The quality of these figures is insufficient. This should be corrected.
8. Please, provide error bars for each data point in Figure 10.
9. It is stated that “On the most surface of the coating, the microhardness is relatively stable in 550 HV. Micro hardness at the secondary surface reached the maximum value of 745 HV (0.3 mm from the coating surface), which is about 3.5 times higher than the substrate hardness. This is due to the fact that the molten reservoir rises from the coating surface during solidification by cooling and then with oxidation. It creates a loose coating on the surface, which reduces the hardness of the coating”. It turns out that the protection against oxidation was insufficient. This should be explained.
10. It is necessary to discuss the results in more detail, the discussion looks incomplete.
11. Conclusions: The conclusions look fragmentary. It is necessary to improve significantly the conclusions so that they fully reflect the results obtained, and, most importantly, clearly show what the main achievement is.
Reviewer 2 Report
In this study, an effort was made to enhance the mechanical properties of Cr12MoV by laser cladding Ni60 alloy reinforced by WC and the outcome oft experimental results were discussed. Although, the experimental methods and analysis is appealing, the manuscript needs major changes. Once these deficiencies are fixed, the manuscript is ready for publication. Finally make the English usage more academic.
1. Typo in Cr12MoV, Cr12, Ni60
2. Replace the following sentence
hardness of the coating reached 745 HV, is 3.5 times
with
hardness of the coating reached 745 HV, which is 3.5 times
3. Why Cr12MoV was selected as the substrate?
4. Space related issue: 10 mm×10 mm×5 m
5. Figure 5 (a) is not clear.
6. Figure 6 (a, b) is not clear.
7. Figure 7 (a) is not clear.
8. Figures 11 (a, b) have Chinese words….
9. Figure 12 is not clear.
9. Replace
4. Discussion
With
4. Conclusion
10. In conclusion section, provide a brief paragraph on main theme of your experiment
Such as: in this study, an attempt was made to improve the
mechanical properties of 9 Cr12MoV by laser cladding Ni60 alloy reinforced by WC
Reviewer 3 Report
The manuscript is well written and within the scope of the journal. However, some points listed below need to be clear
1- 2. Materials and Methods
· What are the shape and dimensions of Ni and WC powders?
· How you adjusted the layer height and how did you confirm the height?
2- 2.2. Laser machining
· How the levels of the process parameters (laser power and scanning speed) were chosen is not explained.
3- 3.2.2. Analysis of the microstructure of the coating
· Could you measure the dilution%? Could you correlate the dilution% to your results for more explanation?
· I expected overheating issue at the last tracks, could you investigate it?
· Also, the use of WC is associated with micro-cracks, explain how you avoided it.
4- 3.2.3. Analysis of dendrite change with the increasing of the layer depth
· You set the layer height to 1mm, what is the actual layer height after the cladding process? What is the reason for the difference in layer height before and after the laser cladding process?
5- Fig.4 XRD measurements
· XRD of the powder before the cladding process should be added for comparison
6- Fig 5. (a) Cross-sectional morphology of the coating
· You may add SEM for the interface for a better explanation
Round 2
Reviewer 1 Report
The revised version of the manuscript is acceptable.
